# Anti-MRSA Sesquiterpenes from the Semi-Mangrove Plant *Myoporum bontioides* A. Gray

**DOI:** 10.3390/md16110438

**Published:** 2018-11-08

**Authors:** Li-Mei Dong, Li-Lan Huang, Hang Dai, Qiao-Lin Xu, Jin-Kui Ouyang, Xu-Chao Jia, Wen-Xiang Gu, Jian-Wen Tan

**Affiliations:** 1State Key Laboratory for Conservation and Utilization of Subtropical Agro-bioresources/Guangdong Key Laboratory for Innovative Development and Utilization of Forest Plant Germplasm, College of Forestry and Landscape Architecture, South China Agricultural University, Guangzhou 510642, China; lmdong@scau.edu.cn (L.-M.D.); ouyangjack@scau.edu.cn (J.-K.O.); 2College of Materials and Energy, South China Agricultural University, Guangzhou 510642, China; lnhuang3233@163.com (L.-L.H.); hangdai0101@163.com (H.D.); 3Guangdong Provincial Key Laboratory of Bio-control for the Forest Disease and Pest, Guangdong Academy of Forestry, Guangzhou 510520, China; qlxu@sinogaf.cn; 4Key Laboratory of Functional Foods, Ministry of Agriculture/Guangdong Key Laboratory of Agricultural Products Processing/Sericultural & Agri-Food Research Institute, Guangdong Academy of Agricultural Sciences, Guangzhou 510610, China; jiaxuchao@gdaas.cn

**Keywords:** *Myoporum bontioides*, methicillin-resistant *Staphylococcus aureus* (MRSA), sesquiterpene alkaloids

## Abstract

The striking rise of methicillin-resistant *Staphylococcus aureus* (MRSA) infections has become a serious threat to public health worldwide. In an effort to search for new anti-MRSA agents from natural products, a bioassay-guided phytochemical study was conducted on the semi-mangrove plant *Myoporum bontioides* A. Gray, which led to the isolation of two new sesquiterpene alkaloids (**1** and **2**) and six known furanosesquiterpenes (**3**–**8**). Their structures were elucidated on the basis of extensive analysis of their 1D, 2D NMR and mass spectroscopic data. These two new alkaloids (**1** and **2**) displayed potent anti-MRSA activity with MIC value of 6.25 μg/mL. This is the first report of sesquiterpene alkaloids from the plants of *Myoporum* genus and their anti-MRSA activity.

## 1. Introduction

Methicillin-resistant *Staphylococcus aureus* (MRSA) infections have become a global threat to public health [1,2,3]. MRSA is responsible for several intractable infections in human being including skin and soft tissue infections, septicemia, endocarditis, pneumonia, enteritis, meningitis, osteomyelitis as well as toxic shock syndrome [4,5]. MRSA infections worldwide have increased rapidly from 1–5% in the mid-1980s to 60–70% today since MRSA was first discovered by British scientist Jevons in 1961 [6]. At present, MRSA infection has surpassed hepatitis B and AIDS, ranking the first among the three most intractable infectious diseases throughout the world [7]. In a response to antimicrobial stress, almost all clinical MRSA isolates produce *β*-lactamase and a penicillin-binding protein with low affinity for *β*-lactam antibiotics [8,9]. Although a variety of non-*β*-lactam antibiotics such as vancomycin, teicoplanin, linezolid, and daptomycin had been recommended for the treatment of MRSA infections [10,11,12,13], a series of drawbacks including slow bactericidal activity, low tissue penetration, and increasing reports of resistance were described and greatly restricted their utility [13,14,15,16,17,18,19]. Therefore, there is an urgent need to discover alternative anti-MRSA candidates with novel structure scaffold and mechanism of action for the treatment of infections arising from MRSA.

One strategy to develop new anti-MRSA agents is to search for anti-MRSA substances or lead compounds from natural products, which has been proven to be effective in the field of new drug development [20]. During our ongoing investigation to search for novel antibiotics from traditional Chinese medicinal plants, *Myoporum bontioides* A. Gray was found to possess anti-MRSA activity. *M. bontioides*, belonging to the genus *Myoporum* in the family Myoporaceae, is a semi-mangrove plant distributed mainly in China, Japan, Australia, New Zealand, Mauritius, and the Hawaiian Islands [21,22]. It grows above the tide lines by the sea and adapts to saline-alkali sand and rocky land, which plays an important role in wind-breaking and sand-fixation, as well as greening the environment [23]. In China, *M. bontioides* has been used as a folk medicine for antidermatosis, antipyretic, and antipsychotic [24,25,26,27]. Previous phytochemical studies have revealed some structurally diverse chemicals from this plant, including sesquiterpenoids, iridoids, monoterpenes, phenylethanoids, and flavonoids, some of which showed important bioactivities [28,29,30,31,32,33]. Our previous experiment showed that the extract of *M. bontioides* possessed anti-MRSA activity. With the aim to find out the potential anti-MRSA substances from *M. bontioides*, we carried out a bioassay-guided phytochemical study on the semi-mangrove plant *M. bontioides*, which led to the isolation of two new sesquiterpene alkaloids (**1** and **2**) and six known furanosesquiterpenes (**3**–**8**) (Figure 1). Their structures were elucidated on the basis of extensive spectroscopic analysis. Herein, we report the isolation and structure elucidation of these compounds, as well as their anti-MRSA activity.

## 2. Results and Discussion

Compound **1** was obtained as a colorless oil with a molecular formula of C_15_H_23_NO_3_ as determined on the basis of HR-EI-MS data, *m*/*z* 265.1660 ([M]^+^), which required five degrees of unsaturation. The ^1^H NMR spectrum (Table 1) showed the signals of three methyls at *δ*_H_ 0.90 (3H, d, *J* = 6.5 Hz), 0.91 (3H, d, *J* = 6.5 Hz) and 1.27 (3H, s), an oxymethine at *δ*_H_ 4.78 (1H, t, *J* = 7.3 Hz), and an olefinic methine at *δ*_H_ 5.97 (1H, t, *J* = 1.5 Hz). The ^13^C NMR spectrum (Table 1), coupled with HSQC analysis, exhibited the signals of fifteen carbons in total, comprising three methyls, five methylenes, three methines, and four quaternary carbons including one oxygenated quaternary carbon at *δ*_C_ 82.6 (C-7), two carbonyl carbons [*δ*_C_ 174.9 (C-15) and 209.2 (C-9)], and an olefinic carbon at *δ*_C_ 163.8 (C-3). Detailed analysis of the NMR data indicated that compound 1 was similar to (–)-epingaione [34,35], a known furanosesquiterpene which was also obtained in this study as compound 3. The main difference was that the signals of the furan group at C-4 in 3 were absent in 1. Instead, proton and carbon signals of an *α*,*β*-unsaturated butyrolactam moiety in 1 were exhibited. These findings led us to establish the structure of 1 as shown in Figure 1. This assignment was in accordance with the molecular formula of 1 (C_15_H_23_NO_3_) and well supported by the 2D NMR spectroscopic data. The heteronuclear multiple bond correlations (HMBC) (Figure 2) from *δ*_H_ 4.78 (H-4) to 121.1 (C-2), 163.8 (C-3), and 174.9 (C-15), from *δ*_H_ 4.00 (H-1) to 121.1 (C-2), 163.8 (C-3), and 174.9 (C-15), from *δ*_H_ 5.97 (H-2) to 47.7 (C-1), 163.8 (C-3), 76.1 (C-4), and 174.9 (C-15), combining with the ^1^H–^1^H COSY correlation between *δ*_H_ 4.00 (H-1) and 5.97 (H-2) (Figure 2), confirmed the presence of the *α*,*β*-unsaturated butyrolactam moiety. The observation of significant NOE correlation of H-4/H-8 (Figure 2) and the absence of NOE correlation of H-4/CH_3_-14 in the NOESY spectrum further supported the *β*-orientation of H-4 and *α*-orientation of CH_3_-14. Consequently, the structure of compound 1 was elucidated as shown in Figure 1, trivially named as myoporumine A.

Compound **2**, obtained as a colorless oil, was deduced to have the molecular formula C_15_H_23_NO_2_ by the HR-EI-MS data, *m*/*z* 249.1720 ([M]^+^), which required five degrees of unsaturation. Its ^1^H and ^13^C NMR spectra (Table 1), in combination with HSQC analysis, indicated three methyls, five methylenes, three methines and four quaternary carbons [including two carbonyl carbons at *δ*_C_ 176.8 (C-15) and 203.7 (C-9), two olefinic carbons at 139.7 (C-3) and 159.8 (C-7). A detailed comparison of the NMR data (Table 1) revealed that compound **2** closely resembled **1** with the main differences of the absence of the oxygen bridge between C-4 and C-7, and the presence of the double bond between C-7 and C-8. This deduction was in accordance with the molecular formula of **2** (C_15_H_23_NO_2_) and well supported by the 2D NMR spectroscopic data, including HSQC, ^1^H−^1^H COSY, and HMBC data. The ^1^H–^1^H COSY correlations of H-1/H-2, H-4/H-5, H-5/H-6, H-10/H-11, H-11/H-12, and H-11/H-13, together with the HMBC correlations from H-4 to C-2, C-3, and C-15, from H-14 to C-6, C-7, C-8, and C-9, and from H-8 to C-6, C-7, C-9, C-10, and C-14, supported the above deduction (see Figure 3). Hence, the structure of compound **2** was determined as shown in Figure 1, trivially named as myoporumine B.

The six known compounds (**3**–**8**) were identified as (−)-epingaione (**3**) [34,35], (–)-dehydroepingaione (**4**) [36], myoporone (**5**) [37], dehydromyoporone (**6**) [37], 9-(3-furanyl)-2,6-dimethyl-4-nonanone (**7**) [38] and dihydrocrassifolone (**8**) [39] respectively, by comparing their spectroscopic data with those reported in the literature.

All the isolated compounds (**1**–**8**) were evaluated for their anti-MRSA activity using the microdilution method as we described previously [40]. As shown in Table 2, these two new alkaloids (**1** and **2**) displayed potent anti-MRSA activity with MIC value of 6.25 μg/mL.

## 3. Experimental Section

### 3.1. General Experimental Procedures

1D and 2D Nuclear magnetic resonance (NMR) spectra were recorded on a Bruker DRX-500 NMR spectrometer (Bruker Biospin Gmbh, Rheistetten, Germany). High-resolution (HR) EI–MS was obtained on a Waters AutoSpec Premier P776 mass spectrometer (Waters, Milford, MA, USA). UV spectra were acquired on a Perkin-Elmer Lambda 650 UV–vis spectrometer (Perkin-Elmer, Inc., Waltham, MA, USA). Optical rotations were measured on a Perkin-Elmer Model 341 polarimeter (Perkin-Elmer, Inc., Waltham, MA, USA). Column chromatography (CC) was performed with silica gel (80–100 mesh, Qingdao Haiyang Chemical Co., Qingdao, China), Sephadex LH-20 (Pharmacia Fine Chemical Co. Ltd., Uppsala, Sweden). Preparative HPLC was performed with an HPLC system equipped with a Shimadzu LC-6AD pump and a Shimadzu RID-10A refractive index detector using a Shim-pack PRC-ODS C-18 column (5 μm, 20 × 250 mm). Thin-layer chromatography (TLC) was conducted on precoated silica gel plates (HSGF254, Yantai Jiangyou Silica Gel Development Co. Ltd., Yantai, China) and spot detection was performed by spraying 10% H_2_SO_4_ in ethanol, followed by heating. Analytical grade chloroform, methanol, petroleum ether, acetone, *n*-hexane, and ethyl acetate were purchased from Tianjin Fuyu Fine Chemical Industry Co. (Tianjin, China).

### 3.2. Plant Material

Leaves of *M. bontioides* were collected from the Leizhou Peninsula, Guangdong province, China in September 2010, and identified by Prof. Bingtao Li from South China Agricultural University. A voucher specimen (No. 20100915) was deposited in the College of Materials and Energy, South China Agricultural University.

### 3.3. Extraction and Isolation

The air-dried leaves of *M. bontioides* (12 kg) were powdered and extracted by supercritical CO_2_ extraction technology at 15 MPa and 30 °C for 15 min yield a supercritical CO_2_ extract (116 g). The crude extract was subjected to silica gel column chromatography, eluted with petroleum ether/acetone (from 100:0 to 0:100, *v*/*v*), to afford fractions F_1_–F_6_ after pooling according to their TLC profiles. According to the result of activity screening, Fraction F_4_ (2.02 g) showed the most potent anti-MRSA activity with MIC value of 25 μg/mL. Then it was subjected to silica gel column chromatography with the elution of chloroform/methanol (from 100:1 to 100:10) to provide subfractions F_4-1_–F_4-4_. Subfraction F_4-1_ was further chromatographed over a silica gel column eluting with *n*-hexane/ethyl acetate (20:1 and 15:1) to afford compounds **5** (15 mg), **6** (10 mg), and **7** (4 mg). Subfraction F_4-2_ was separated by Sephadex LH-20 column chromatography eluted with acetone to give compounds **3** (6 mg) and **4** (4 mg). Subfraction F_4-3_ was recrystallized to give compound **8** (5 mg). Subfraction F_4-4_ was applied on Sephadex LH-20 column chromatography with the elution of chloroform/methanol (1:4, *v*/*v*) to give F_4-4-1_ and F_4-4-2_. F_4-4-1_ was further purified by preparative HPLC with a Shim-pack PRC-ODS C-18 column (5 μm, 20 × 250 mm) using 40% methanol in water (*v*/*v*) as mobile phase at the flow rate of 8 mL/min to obtain compound **1** (4 mg, *t*_R_ 65 min). F_4-4-2_ was further purified by preparative HPLC using 25% acetonitrile in water as mobile phase at the flow rate of 10 mL/min to yield compound **2** (3 mg, *t*_R_ 58 min).

Myoporumine A (**1**): colorless oil; [α${]}_{D}^{20}$–14.5 (*c* 0.20, CHCl_3_); UV (CHCl_3_) λ_max_ nm (log *ε*) 255 (3.37); HR-EI-MS: *m*/*z* 265.1660 [M]^+^ (calcd 265.1678, C_15_H_23_NO_3_); ^1^H NMR and ^13^C NMR data, see Table 1. The NMR and HREIMS spectra, see Appendix A.

Myoporumine B (**2**): colorless oil; UV (CHCl_3_) λ_max_ nm (log *ε*) 230 (3.21), 255 (3.47); HR-EI-MS: *m*/*z* 249.1720 [M]^+^ (calcd 249.1723, C_15_H_23_NO_2_); ^1^H NMR and ^13^C NMR data, see Table 1. The NMR and HREIMS spectra, see Appendix A.

### 3.4. Anti-MRSA Assay

The anti-MRSA activity of compounds **1**–**8** was evaluated by the microdilution method as we described previously [40]. The MRSA strain (No. 11646) was provided by State Key Laboratory Respiratory Disease, Guangzhou Institute of Respiratory Disease (Guangzhou, China), which was resistant to methicillin and sensitive to vancomycin. Resazurin was used as a visible indicator in the assay. The count of bacterial suspension was adjusted to 1 × 10^5^ CFU/mL with MHB. Test samples were diluted with the medium (DMSO) by the two-fold dilution method. The final concentrations of each sample in the wells were 100, 50, 25, 12.5, 6.25, 3.12, 1.56, and 0.78 μg/mL. Vancomycin was used as a positive control. Finally, the plates were incubated at 37 °C for 5–6 h until the color of negative control wells (which contained DMSO instead of the test sample) change to pink. The lowest concentration for each test sample at which color change occurred was recorded as the minimal inhibitory concentration (MIC). MIC values of test samples were displayed in Table 2.

## 4. Conclusions

In summary, two new sesquiterpene alkaloids (**1** and **2**) and six known furanosesquiterpenes (**3**–**8**) were isolated from the semi-mangrove plant *M. bontioides*. Their structures were elucidated on the basis of extensive analysis of their 1D, 2D NMR and mass spectroscopic data. These two new alkaloids **1** and **2** showed potent anti-MRSA activity, suggesting that they could be worthy of consideration for the development and research of anti-MRSA agents. This is the first report of sesquiterpene alkaloids from the plants of *Myoporum* genus and their anti-MRSA activity.

## Figures and Tables

**Figure 1 marinedrugs-16-00438-f001:**
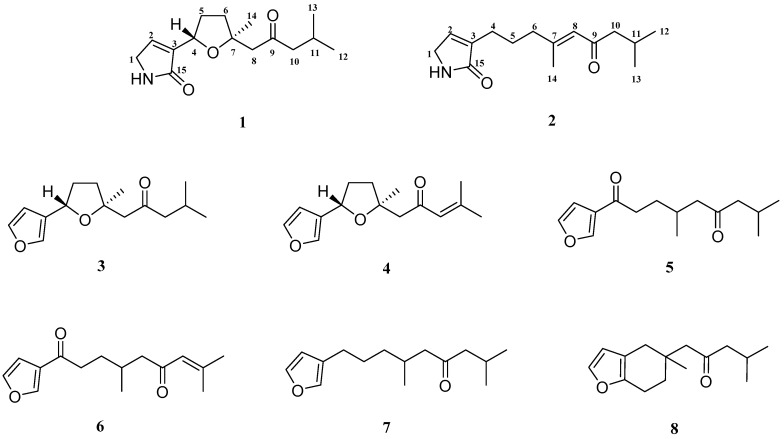
Chemical structures of compounds **1**–**8**.

**Figure 2 marinedrugs-16-00438-f002:**
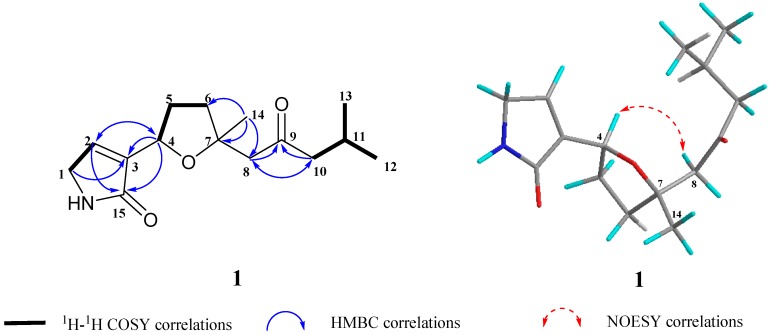
Key ^1^H–^1^H COSY, HMBC and NOESY correlations of compound **1**.

**Figure 3 marinedrugs-16-00438-f003:**
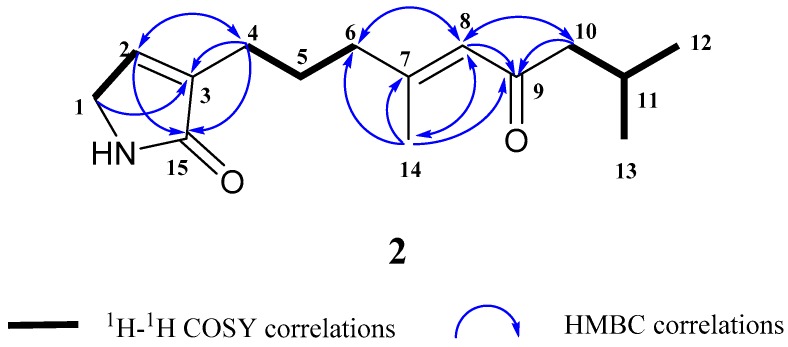
Key ^1^H–^1^H COSY and HMBC correlations of compound **2**.

**Table 1 marinedrugs-16-00438-t001:** ^1^H (600 MHz) and ^13^C (150 MHz) NMR data of compounds **1** and **2** in CDCl_3_.

H/C	1		2	
	*δ*_H_ (mult, *J* in Hz)	*δ* _C_	*δ*_H_ (mult, *J* in Hz)	*δ* _C_
1	4.00 (brs)	47.7 (CH_2_)	3.94 (d, 1.6)	47.8 (CH_2_)
2	5.97 (t, 1.5)	121.1 (CH)	6.94 (t, 1.5)	140.6 (CH)
3		163.8 (C)		139.7 (C)
4	4.78 (t, 7.3)	76.1 (CH)	2.26 (m)	25.9 (CH_2_)
5	1.88 (m) 2.24 (m)	32.5 (CH_2_)	1.77 (m)	26.8 (CH_2_)
6	1.94 (m) 2.09 (m)	37.2 (CH_2_)	2.23 (m)	41.6 (CH_2_)
7		82.6 (C)		159.8 (C)
8	2.67 (q, 15.3)	53.4 (CH_2_)	6.19 (q, 1.1)	124.9 (CH)
9		209.2 (C)		203.7 (C)
10	2.32 (dd, 6.9, 2.5)	53.7 (CH_2_)	2.31 (d, 7.0)	54.3 (CH_2_)
11	2.13 (m)	24.5 (CH)	1.71 (m)	26.4 (CH)
12	0.90 (d, 6.5)	22.7 (CH_3_)	0.92 (d, 6.6)	22.9 (CH_3_)
13	0.91 (d, 6.5)	22.7 (CH_3_)	0.92 (d, 6.6)	22.9 (CH_3_)
14	1.27 (s)	27.6 (CH_3_)	2.12 (d, 1.3)	19.4 (CH_3_)
15		174.9 (C)		176.8 (C)

**Table 2 marinedrugs-16-00438-t002:** In vitro anti-MRSA activity of the Fraction F_4_ and compounds **1**–**8**.

Sample	MIC (μg/mL)	Sample	MIC (μg/mL)
Fraction F_4_	25	**5**	50
**1**	6.25	**6**	50
**2**	6.25	**7**	>100
**3**	25	**8**	>100
**4**	25	Vancomycin	0.78

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
