# Peer review of "Anti-MRSA Sesquiterpenes from the Semi-Mangrove Plant Myoporum bontioides A. Gray"

_marinedrugs, 2018, doi:10.3390/md16110438_

Round 1

Reviewer 1 Report

Authors isolated two new sesquiterpene alkaloids (1 and 2) from plant M. bontioides. They described their structures on the basis of 1D, 2D NMR and mass spectroscopy. These two alkaloids showed anti-MRSA activity. The article is good written, with proper methodology. However, I would like suggest some corrections. Why authors studied only MRSA strains? I think it would be better describe results for both MSSA and MRSA. In the article is lack of information about studied MRSA strain: 1. if only one MRSA strain was tested or many, 2.where did it come from, 3. whether it was a clinical strain or, for example, from ATCC, 4. to which antibiotics this strain was resistant and to which sensitive, 5. did this strain have other resistance mechanisms eg. penicilinase production and/or MLSb? It is possible that the study on several MRSA strains give other results eg. various MICs.

Author Response

Thank you for your valuable suggestions. We have supplemented the information about studied MRSA strain in the “Anti-MRSA Assay” part of this article. 

Since our recently initiated program is aimed at searching for anti-MRSA substances from traditional Chinese medicinal plants, so only MRSA strain was studied in the bioassay-guided phytochemical investigation on Myoporum bontioides A. Gray. 

We will investigate the antimicrobial activity of the isolated compounds against MSSA and other strains in the further work. 

The MRSA strain (No.11646) was a clinical strain from Guangzhou Institute of Respiratory Disease, which was resistant to methicillin and sensitive to vancomycin. 

It is not clear if there are other resistance mechanisms. In future work, we will consider to study several MRSA strains as you suggested.

Reviewer 2 Report

Taking the extending role of MRSA infections on intesive care units into consideration, the identification and characterization of new putataive drugs or lead compounds, respectively, is an important issue in the clinic situation. Therefore the study of Dong et al., identifying 8 possible new compounds to treat MRSA, is of significant interest for the scientific community. I am looking forward to read in a follow up study about the effect of the newely identified compounds in an animal based setup. 

Author Response

Thank you for your beneficial suggestions. We will consider to study the anti-MRSA effect of the newly identified compounds in an animal model as you suggested, if we can get more amount of these isolated compounds in further research.